# Extracellular Vesicles from *Campylobacter jejuni* CDT-Treated Caco-2 Cells Inhibit Proliferation of Tumour Intestinal Caco-2 Cells and Myeloid U937 Cells: Detailing the Global Cell Response for Potential Application in Anti-Tumour Strategies

**DOI:** 10.3390/ijms24010487

**Published:** 2022-12-28

**Authors:** Mariele Montanari, Michele Guescini, Ozan Gundogdu, Francesca Luchetti, Paola Lanuti, Caterina Ciacci, Sabrina Burattini, Raffaella Campana, Claudio Ortolani, Stefano Papa, Barbara Canonico

**Affiliations:** 1Department of Biomolecular Sciences, University of Urbino Carlo Bo, 61029 Urbino, Italy; 2Faculty of Infectious and Tropical Diseases, London School of Hygiene & Tropical Medicine, London WC1E 7HT, UK; 3Department of Medicine and Aging Science, “G. d’Annunzio” University of Chieti-Pescara, 66100 Chieti, Italy

**Keywords:** *C. jejuni* CDT, EVs, EV uptake, anticancer strategies, flow cytometry

## Abstract

Cytolethal distending toxin (CDT) is produced by a range of Gram-negative pathogenic bacteria such as *Campylobacter jejuni*. CDT represents an important virulence factor that is a heterotrimeric complex composed of CdtA, CdtB, and CdtC. CdtA and CdtC constitute regulatory subunits whilst CdtB acts as the catalytic subunit exhibiting phosphatase and DNase activities, resulting in cell cycle arrest and cell death. Extracellular vesicle (EV) secretion is an evolutionarily conserved process that is present throughout all kingdoms. Mammalian EVs play important roles in regular cell-to-cell communications but can also spread pathogen- and host-derived molecules during infections to alter immune responses. Here, we demonstrate that CDT targets the endo-lysosomal compartment, partially evading lysosomal degradation and exploiting unconventional secretion (EV release), which is largely involved in bacterial infections. CDT-like effects are transferred by Caco-2 cells to uninfected heterologous U937 and homologous Caco-2 cells. The journey of EVs derived from CDT-treated Caco-2 cells is associated with both intestinal and myeloid tumour cells. EV release represents the primary route of CDT dissemination, revealing an active toxin as part of the cargo. We demonstrated that bacterial toxins could represent suitable tools in cancer therapy, highlighting both the benefits and limitations. The global cell response involves a moderate induction of apoptosis and autophagic features may play a protective role against toxin-induced cell death. EVs from CDT-treated Caco-2 cells represent reliable CDT carriers, potentially suitable in colorectal cancer treatments. Our data present a potential bacterial-related biotherapeutic supporting a multidrug anticancer protocol.

## 1. Introduction

### 1.1. Significance Statement

Bacteria, particularly bacterial toxins, can be applied in cancer therapy, underscoring both the benefits and limitations. To investigate the effects of *Campylobacter jejuni* CDT on intestinal and myeloid tumour cells, we created a reference grid to compare data obtained through the administration of EVs from *C. jejuni* CDT-treated Caco-2 cells. These entities demonstrated antiproliferative effects on tumour cells as well as the alteration of mitochondrial-lysosomal axis. All of these events can indicate a novel strategy for anticancer treatment.

### 1.2. CDT: Structure, Function, and Its Potential Applications

CDT is a heterotrimeric holotoxin encoded by a range of Gram-negative pathogenic bacteria [1]. Human pathogens that produce CDTs include *Campylobacter jejuni*, which is responsible for human enterocolitis [2,3]. CDT consists of three subunits, A, B, and C (23, 29, and 21 kDa, respectively), encoded from the cluster of *cdtA, cdtB,* and *cdtC* genes. CDT toxicity is primarily dependent on the CdtB nuclease activity [4,5,6]. CdtA and CdtC act as carriers to transport the catalytic subunit, CdtB, within the host cells [7]. Inside the nucleus, CdtB exhibits a DNase I type activity capable of triggering damage to DNA [4]. These cause the activation of DNA repair complexes by inducing a blockage of the cell cycle in G2/M before cell division [5,8,9]. In the eukaryotic cell, initially, there will be a cytoplasmic distension followed by cell death through mechanisms not yet elucidated [5,8,9,10]. The CDT toxin exerts multiple effects such as (i) the release of IL-8 by intestinal epithelial cells [11]; (ii) the promotion of DNA repair in the host cell [12]; (iii) gastroenteritis in NF-kB deficient mice [13] as well as (iv) cell death of human monocytes [14,15]. In humans, the symptoms of disease could be due to CDT-induced cell death and the resulting inflammatory response [16,17]. In previously published manuscripts [8], we demonstrated that after CDT-containing lysate treatment, HeLa cells increased their endolysosomal compartment because of toxin internalisation, with an endosome-lysosome volume/number increase, exploiting the GPI-Ps pathway of endocytosis. In addition, human monocytes [15] revealed relevant mitochondrial alterations, confirming an intrinsic apoptotic pathway and changes in lysosomal acidic compartments and p53 expression due to CDT action. Finally, in U937 myeloid tumour cells, we also investigated CD317/tetherin, an organiser of membrane microdomains [9]. This is an antigen localised within lipid rafts on the cell surface in the TGN and/or within recycling endosomes, and is increased after active CDT-containing lysate treatment [18]. The effects of the toxin vary depending on the type of eukaryotic cell infected. Generally in humans, CDT can inhibit both humoral and cellular immunity by inducing apoptosis in immune cells as well as necrosis in the epithelial cells of the intestine [19,20]. Therefore, CDT has the potential to be used as an anti-tumour agent.

Currently, cancer therapy employs all subunits of CDT, or only the enzymatic subunit CdtB in combination with some targeted portions. Bachran et al. [21] reported that conjugation of CdtB of *Haemophilus ducreyi* with the N-terminal 255 amino acids of *Bacillus anthracis* toxin lethal factor has the potential to act as an anticancer treatment. Additionally, Chen and colleagues utilised nanoparticles based on hyaluronic acid for the delivery of CdtB for prostate cancer treatment [22]. The authors demonstrated that this composition has activity like the CDT whole toxin, but with the difference that the CdtB can be delivered specifically to cancer cells and enhance the effect of ionising radiation in radio resistant prostate cancer cells [22]. Additionally, Vafadar et al. [23] demonstrated new immune-toxins based on single-chain variable fragment associated with CDT (ScFv-CdtB) that were designed and evaluated against breast cancer. Keshtvarz et al. produced an engineered version of CdtB to reduce immunogenicity and maintain stability as a new drug candidate for tumour therapy [24]. The authors identified hot spot immunogen regions using computational methods followed by the mutation of these regions to eliminate B-cell epitopes. Computational approaches can be cost-effective methods to reduce the immunogenicity of the engineered therapeutic proteins, however, experimental studies are crucial to confirm the results. The active CDT can be released in bacterial extracellular vesicles (EVs), known as outer membrane vesicles (OMV) [25,26,27]. *C. jejuni* produces OMVs, which carry a range of proteins where some are involved in the delivery of virulence factors to infect host cells and can be exploited for the dissemination of CDT [27]. 

### 1.3. Extracellular Vesicle Features and Their Role in Host–Bacteria Interactions

EV secretion is an evolutionarily conserved process present throughout all kingdoms [28]. EVs are known as cell-derived membranous structures that can transport various active biomolecules from creator cells to target cells, thereby changing the physiology of the target cells [29,30,31]. EVs have been isolated from various sources including mammalian and prokaryotic cell cultures, blood plasma, bovine milk, and plants [28]. EVs make up a heterogeneous population of particles that are generally classified into three distinct sub-populations based on their biogenesis: microvesicles, apoptotic bodies, and exosomes [31]. Microvesicles arise from the direct outer budding of the plasma membrane, producing a population of EVs that are heterogeneous in size. Apoptotic bodies are also generated from the cell surface, though they are only released by dying cells during cell fragmentation [32,33]. Exosomes are formed from inward budding of the limiting membrane of endosomes to form multivesicular bodies (MVBs). Subsequently, exosomes are released into the extracellular space by fusion of MVBs with the plasma membrane. Following release from the cell surface, exosomes can interact with the extracellular matrix or elicit a response in cells within the microenvironment or at a distance. Exosomes have a size ranging from 40 to 120 nm, while microvesicles exhibit a size of 50–1000 nm [34,35]. Due to their overlapping sizes, surface markers, and the absence of proteins that are restricted to specific populations, it remains a challenge to distinguish exosomes and microvesicles. Thus, all different types of vesicles are referred to as EVs. Moreover, EVs may have the capacity to cross biological barriers [36,37,38], exploit endogenous intracellular trafficking mechanisms, and trigger a response upon uptake by recipient cells [39].

EVs contain numerous biomolecules such as proteins, miRNAs, mRNAs, long non-coding RNAs, DNA strands, lipids, and carbohydrates from parental cells, which when delivered to target cells are involved in the reprogramming of cell fate, with modified functionality and morphology [40,41]. EVs play critical roles in the progression of different pathological conditions [42]. EVs from infected cells contain virus particles that induce virus infection in healthy cells and modulate immune responses of the host [43,44]. EVs released by host cells infected with *Helicobacter pylori* contained the virulence factor CagA and could reach macrophages that contribute significantly to inflammation, and in doing so, promote the development of disease [45]. Such abilities have drawn a great deal of attention towards EVs as a therapeutic application and as a prospective vehicle for the delivery of therapeutics that could overcome issues related to liposomes and other synthetic drug delivery systems [31,46]. 

This study is divided into two parts. We initially focused on the direct effects of the toxin-containing lysates in intestinal Caco-2 and U937 myeloid cells, investigating lethal and sublethal alterations such as cell cycle, blocking, mitochondrial modifications, ROS increase, relevant changes in lysosomal exocytosis, secretory autophagy, and EV release. The findings emerging from this part of our study represent the foundation for our second part, which is to investigate the presence of an active CDT in EVs produced and released by CaCo-2 treated cells. This topic was addressed, revealing the effects exerted in the two different tumour recipient cell lines U0937 and Caco-2. The role of EVs from *C. jejuni* Caco-2 infected cells in modulating tumour epithelial and myeloid response is, for the first time, highlighted by the data presented herein. These EVs function as a carrier of antiproliferative signals in our in vitro models. 

## 2. Results and Discussion

### 2.1. PART I: Building the Reference Grid for Cell Response to CDT and EV Characterisation

#### Flow Cytometric Analysis of Viability and Mitochondria Health: Transmembrane Potential and ROS Generation

Our previous research on *C. jejuni* toxin effects on different cell lineages (Hela, monocytes and U937) [8,9,15] demonstrated that a significant and relevant cell death was induced after 72 h. Our cell death data demonstrated that the wild-type lysate caused a significant increase in 7′AAD-positive cells at 72 h in both the Caco-2 and U937 cell lines (Figure 1A,B). Contemporary lysosomal and mitochondria functions are altered, starting at 24–48 h. Therefore, we checked the perturbations of the mitochondrial transmembrane potential on cells treated with the wild-type *C. jejuni* ATCC 33,291 lysate (indicated as ATCC) and *C. jejuni* 11168H cdtA mutant lysate (indicated as MUT) compared to the untreated control cells (indicated as CTRL) at different time-points (from 24 h to 72 h). Mitochondrial functions were evaluated with the mitochondrial membrane potential (MMP) dye, TMRE. Caco-2 cells treated with the wild-type strain displayed elevated TMRE fluorescence intensities with respect to the untreated and the cdtA mutant strain treated cells. These results indicate that *C. jejuni* ATCC-lysates induce mitochondrial hyperpolarisation (Figure 1C,D). Such instabilities of MMP have been described in previous studies [47,48,49,50], and have often been attributed to the oscillations of the mitochondrial permeability transition (MPT) [51], which in a significant portion of cases are associated with ROS increase and oxidative stress. Thus, in Caco-2 cells, we further assessed whether mitochondrial ROS (evaluated by MitoSOX Red) were increased by *C. jejuni* ATCC-lysates. We found that ATCC increased MitoSOX Red positive cells at 48 h and 72 h (Figure 1E). In addition, we investigated intracellular ROS levels, particularly hydrogen peroxide (Figure 1F). Our data demonstrate that active CDT specifically targets mitochondria, inducing ROS generation, whereas the mutant lysate increased, to a lesser extent, the intracellular ROS level (Figure 1).

### 2.2. Lysosomal and Autophagic-Like Vacuoles Involvement and Aberrant Endocytic Activity Detection

Our investigation on lysosomes, the autophagic pathway, and differentiation of endo-lysosomal vacuoles was conducted through the markers LysoTracker Deep Red (LTDR), monodansylcadaverine (MDC), Acridine Orange (AO), and LAMP-1.

Our data demonstrated autophagic-like vacuole accumulation in the ATCC CDT treated cells (Figure 2A,B). However, MDC cannot entirely describe macroautophagy because it also marks autophagic compartments after their fusion to acidic endo/lysosomes.

Furthermore, we assessed the rate of acidic vesicular organelle accumulation using AO. This metachromatic probe can evaluate both more acidic (FL3) and less acidic (FL1) vacuoles. These are usually referred to as early endosomes, since they are still not mature and are acidified to pH 6.2 by V-ATPase [15,52]. The ratio FL3/FL1 describes this compartment. Data from this acidotrophic dye (Figure 2C) not only confirmed the results obtained by LTDR, but also suggests the modulation of conventional and unconventional secretion. These finding are confirmed by the quantification of the surface expression of LAMP-1/CD107a (Figure 2D), a major integral membrane glycoprotein of late endosomes and lysosomes. Moreover, the LTDR data showed a slightly progressive increase in lysosomes from 24 h to 48 h for the cells treated with the wild-type toxin (Figure 2E,F). Confocal analyses detail the expansion of the lysosomal compartment and their massive positioning in the periphery (Figure 2E).

Therefore, we detected the involvement of lysosomal exocytosis and secretory autophagy. The autophagic machinery, particularly those vesicular components here investigated, is a fundamental mechanism for toxic protein disposal, immune signalling, and pathogen surveillance. However, if the various molecules are not degraded into lysosomes, the mechanism cannot be properly defined as autophagic [40]. Our data are in agreement with Greene et al. [53], who strongly suggested the extracellular release of toxin and bacterial molecules that were previously internalised. Therefore, the effect on neighbouring cells of a quantity of toxic materials released in the extracellular space was investigated.

### 2.3. Lysosomal Exocytosis, EV Release, and Secretory Autophagy: The Autophagic- and Endo-Lysosomal Systems Go Extracellular

Recent evidence has underlined that lysosomal activity plays an important role in the secretion of EVs and the sorting of their cargo. The release of EVs often serves as an alternative disposal pathway to the lysosome, as previously described [54,55]. The EVs produced by intestinal Caco-2 cells treated with *C. jejuni* lysates were evaluated by flow cytometry (FC). In Figure 3A, EVs positive for the LCD probe (lipophilic cationic dye, LCD+) are represented by the red events in the contour plots. By means of the anti CD63 antibody, it is possible to specifically detect exosomes (blue area) as CD63 + vesicles (Figure 3A). Counting beads (green area) allowed for the absolute counting of EVs, highlighting an increase in LCD+ events (EVs) and CD63+ events (exosomes) in the medium of cells treated by the ATCC lysate (Figure 3B,C).

Furthermore, we detected the surface expression of CD107a (LAMP-1) employed in cells to reveal lysosomal exocytosis [56]. In fact, some authors have described the presence of this molecule in the cells (in the endo-lysosomal compartment) [43], while others have also confirmed a possible lysosomal origin of the vesicles [57]. In fact, these EVs show a larger dimension in respect to classic exosomes; for this reason, we enumerated these EVs independent of CD63 expression. Our data highlight a relevant and significant increase in CD107a positive (CD107a+) EVs in the medium from *C. jejuni* ATCC-lysate-treated Caco-2 cells (Figure 3D). EVs from Caco-2 cells treated by the cdtA mutant only showed a moderate increase, although statistically significant. These findings demonstrate the involvement of the lysosomal pathway and autophagy secretion induced by both lysates (wild type and mutant) to a different extent, whereas the targeting of the endosomal pool and the mechanisms of extracellular release are mainly primed and hijacked by ATCC’s active CDT. These cellular processes underline the crosstalk between the autophagic and the endosomal system and indicate an intersection between degradative and secretory functions, particularly triggered by active CDT.

To enlighten the presence of inflammatory signals inside EVs, we investigated the expression of the selected inflammation-related miRNAs (miR-16 and miR-146a) (Figure 3E,F). These short RNAs are considered important regulators of cytokine gene expression, acting either as posttranscriptional regulators or as repressors of mRNA-translation. Specifically, earlier observations on immune cells from Bhaumik et al. [58] demonstrated that miR-146a/b negatively regulates NF-kB activity and the inflammatory pathway in breast cancer cells [59,60]. Levels of miR-146a in Caco-2-secreted EVs are moderately decreased in response to the exposure to active CDT (Figure 3E). A similar pattern of miR-146a expression was also observed by other researchers [61,62]. Moreover, several studies have revealed that miR-16 could inhibit LPS-induced inflammation [63]. Indeed, You et al. reported findings supporting the role of miR-16 as a tumour suppressor in colorectal cancer cells (CRCs) by targeting KRAS [64]. MiRNA-16 was greatly upregulated in ATCC-treated Caco-2-derived EVs but not in the mutant-treated cells (Figure 3F), demonstrating that the active toxin is a key-factor priming this miRNA loading into EVs. In this regard, it is important to consider both EVs (LCD+ events) and exosomes (CD63+ events).

### 2.4. CDT Effects on Caco-2 Cells and U937 Cells: Evaluation of Cell Cycle Blocking

The efficacy of bacterial lysates was tested by the analysis of cell cycle arrest (typical of CDT). Moreover, the G2/M blocking activity of CDT was here considered as a desirable support in anticancer strategies, and it has been targeted in EV activities, which is highlighted in part II of this study. Briefly, cells were ethanol-fixed and then stained with propidium iodide (PI) at 48 h. As shown in Figure 4A,B, the wild-type lysate induced a cell cycle block in the G2/M phase in Caco-2 cells, whereas this did not take place in the mutant-treated cells. However, aside from this well-known effect, cells treated by the *C. jejuni* lysates showed an increased presence of proliferating cell nuclear antigen (PCNA) in G2/M cells (Figure 4C), in agreement with He et al. [65] and Benzine et al. [66]. As performed previously for U937 cells [9], Caco-2 cells were also tested for pRb expression. Figure 4D shows an overexpression of pRb in cells preincubated with the wild-type lysates after 72 h compared to the control and the cdtA mutant-treated cells. The retinoblastoma protein (pRb) pathway plays a key role in the regulation of several cellular processes, and this protein as well as its regulators (cyclins, cyclin-dependent kinases (Cdks), and Cdk inhibitors) are frequently deregulated in human cancer [67,68]. In quiescent cells, pRb represses the transcription of genes required for DNA replication or mitosis. Its targeting is evident in wild-type ATCC CDT-treated Caco-2 cells in replicative stress. Indeed, pRb is implicated in cellular senescence, together with p53, a protein involved in both senescence induction and maintenance [69]. The abrogation of the cell cycle checkpoint proteins P53 and Rb leads to an unscheduled S phase entry, which can lead to replication stress, DNA damage response, and carcinogenesis. This is another advantageous point to be exploited in antiproliferative/anticancer conditioning protocols.

Host response to CDT-treatment differs depending on the target cells; apoptosis was reported in the T- and B-cell lines whilst other cells preferentially arrest in the G1 and/or G2 phases [70,71,72]. U937 data have shown that *C. jejuni* CDT inhibits proliferation (Figure 4E,F) and the expression of NF-kB p65 in U937 cells (Figure 4G). These findings (the induction of autophagy and cell cycle arrest) correlate with the anticancer effects of several molecules, together with the induction of autophagy [73]. In fact, the NF-kB signal transduction pathway is considered an important pathway that regulates the proliferation and tumourigenesis of several types of cancers [74].

### 2.5. PART II: Detecting the Global Cell Response Induced by EVs and Their Application as Antiproliferative Signal Carriers

#### Co-Culture Results: Subcellular Response of the Co-Cultured Cell Line

A graphical description of the co-culture set-up between the Caco-2 adherent cells/U937 floating cells is presented in Appendix A. Cells were analysed by FC, confocal microscopy (CM), and scanning electron microscopy (SEM). Results from the SEM observations of the Caco-2 cells showed a significant increase in cell size (a typical CDT effect) and a dense network of long apical microvilli on the surface of ATCC-treated cells (Appendix A). The data depicted in Appendix A also showed a decrease in the amount of surface CD133 in the Caco-2 cells, treated with the wild-type CDT (Appendix A). Confocal images of the detached Caco-2 cells highlight the CD133 distribution on the cell surface and on the membrane protrusions as detectable in the white box (white arrow) (Appendix A).

CD133 (prominin-1) is reported to be highly expressed in Caco-2 cells, indicating their features of cancer stem cells (CSCs) [75,76,77]. It is predominantly located on plasma membrane processes and microvilli, which indicates a possible involvement of this molecule in membrane structure organisation. It is involved in the organisation of plasma–membrane protrusions, maintenance of the apical–basal polarity of epithelial cells, biogenesis of the photoreceptive disc, and is a mechanism of multidrug resistance along with the capacity for self-renewal and tumour formation [78,79,80]. In epithelial cells, CD133+ EVs bud as microvesicles/exosomes from primary cilium and/or microvilli present at the apical (Appendix A). Appendix A illustrates that the presence of toxins increases the EV release for both cell lines [81] and specifically for CaCo-2 cells [82,83,84]. Kang et al. demonstrated that the membrane glycoprotein CD133 acts as a novel regulator of EV [85]. In U937 co-cultured with the ATCC-Caco-2 cells-, S, and G2/M phases [86] significantly increased (Appendix A), revealing a proliferation block, normally referred to as DNA injury with cytolethal distending toxins [87]. As an appropriate control for the specific CDT effects, Mut co-cultured U937 samples displayed a cell cycle similar to that from the U937 cells co-cultured with Caco-2 uninfected cells. These data on cell cycle phases and cell death undoubtedly trace the effect of CDT, transferred from intestinal to myeloid cells, pointing out the transfer of the antiproliferative action. In fact, CDT causes DNA damage [66] that leads to two types of cell cycle arrest: G1 arrest and G2/M arrest.

### 2.6. Extracellular Vesicles from Caco-2 Infected-Cells on Homologous Caco-2 and Heterologous U937 Cells

We specifically evaluated the toxin effects mediated by EVs released from infected Caco-2 and transferred to uninfected homologous Caco-2 and heterologous U937 cells (Figure 5A). Briefly, EVs from the medium of Caco-2 cells treated with *C. jejuni* ATCC 33,291 lysate (EvA), *C. jejuni* 11168H cdtA mutant lysate (EvM), and the untreated control cells (EvC) were isolated by ultracentrifugation and characterised using nanoparticle tracking analysis (NTA) together with a flow cytometry approach. EVs labelling by PKH67 allowed us to trace them by FC and CM before and after their interaction with the Caco-2 and U937 cells, respectively. Our data highlight the efficient fluorescence detection (Figure 5B,C) and monitoring of the size and concentration differences in EvA vs. EvM and EvC (Figure 5D,E).

### 2.7. EV Uptake and Evaluation of Specific CDT Effects

To quantitatively evaluate the cellular uptake of EVs, Caco-2 and U937 cells were incubated with 2 × 10^4^ PKH67-labeled EVs/cell for 48 h and analysed by FC and CM (Figure 6). The cell death results after 48 h demonstrated that EvA caused a significant increase in apoptotic cells in both the Caco-2 and U937 cell lines (Figure 6A,B). Our data highlighted a similar EV uptake for EvC and EvA for the Caco-2 cells, whereas EvM appeared less internalised into cells (Figure 6C,D). Indeed, the CM analyses underlined a discrepant intracellular distribution: punctate and scattered in EvC samples, perinuclear and “grouped” in EvA samples (Figure 6E). Of note, after 48 h, data on the EvA-treated samples revealed a significant reduction in absolute cell counting, whereas the EvM cells registered the same count or even more than the EvC cells (Figure 6F). Using FC and CM, we monitored the EV intracellular uptake in the U937 cells (Figure 6I,J). Our data underline a similar EV uptake for EvC and EvA whereas the EvM samples revealed a reduced internalisation in both cell lines (Appendix A), as observed in the EV-treated Caco-2 cells. Moreover, in myeloid cells, the EvA samples only showed a moderate reduction in the cell absolute counts (Figure 6K).

Of note, almost all functional parameters (mitochondrial membrane potential and lysosomal compartment) (Figure 6G,H) were in agreement with those collected from the CDT directly-treated Caco-2 cells; this behaviour (Figure 6L,M) was also evident for the directly-treated U937 cells [9].

By internalising the PKH67 + EVs samples co-stained with LTDR and TMRE (Figure 7), a different subcellular distribution and the typical CDT cell distension was observed, starting at 48 h. Confocal analyses demonstrated that the EVs, specifically in the case of Caco-2 cells-EvA treated, were distributed in the perinuclear region, as shown in Figure 7A. These results are in agreement with Mantel et al. [88], who observed perinuclear localisation of RBC-derived EVs in bone marrow endothelial cells. Lombardo et al. [89] also observed internalisation of endothelial-derived EVs by endothelial cells. EVs can colocalise within lysosomes, indicating that EV-material is directed to the lysosomal network [90]. As observable (Figure 7A,B,E,F), EvA only partially co-localised with lysosomes in both cell lines, probably reflecting the registered decreased effect on mitochondria. Moreover, data on the mitochondrial co-localisation showed a negligible Pearson’s coefficient, similarly for cells + EvC, EvA, and EvM (Figure 7C,D,G,H) excluding the mitochondrial route from the EV intracellular journey. The main targeting of EVs to lysosomes involves the EV destination to late endosomes and lysosomes and/or to early endosomes that subsequently undergo maturation. The process of the fusion of a fraction of internalised EVs with lysosomes, resulting in EV cargo exposure to the cell cytosol, is a mechanism well-described by Joshi et al. [91]. Different sites for EV cargo release in acceptor cells have been proposed including (i) the plasma membrane [92]; (ii) the endosome [93,94]; and (iii) the endoplasmic reticulum [95]. Our experiments revealed that the EV membrane does not remain in the plasma membrane of Caco-2 cells (Figure 6E); in fact, green fluorescence of PKH67-labeled EVs is mainly located in the perinuclear area (where the perinuclear pool of ER accumulates) and is scattered in the cytoplasm, partially co-localising with LTDR, highlighting the targeting of endo-lysosomal systems. With the aim to evaluate the typical CDT-actions, we proceeded to detect the demonstrated mechanistic effects, particularly the blocking of cell proliferation. The study of the fate of EV is technically challenging due to the to the low quantities of encapsulated cargo molecules [91,96].

To evaluate the G2/M block, both cell lines were ethanol-fixed and tested for PKH26-EVs fluorescence, then separately monitored for DNA content. EvC (EVs from uninfected Caco-2 cells) cells showed an uptake similar to the EvA cells (EVs from ATCC infected Caco-2 cells). The gating strategy for the detection of cells with internalised EVs (EV+) is depicted in Appendix A. The two different subsets (EV+ and EV-) were analysed at 72 h to better discriminate the alterations in the cell cycle phases (Figure 8) in both the Caco-2 and U937 cells.

FM analysis was performed to establish whether EvA induced the typical cell cycle arrest, detecting the G2/M block in cells with detected PKH67 EV+ (Figure 8A,B,E,G) and in cells with undetected PKH67 EV− (Figure 8 C,D,F,H). Indeed, the EvA+ and EvM+ subsets highlight an even greater accumulation in the S and G2/M phases in Caco-2 cells (Figure 8B,D). As performed on IECs, the U937 DNA content was evaluated after 72 h from EV administration. The PKH26 EV+ fraction of the EvA U937 samples demonstrated the highest number of G2/M events (Figure 8E,G). Indeed, the cell cycle profiles must be coupled with the absolute cell counts (Figure 6F, Caco-2 cells and M-U937 cells), clearly describing an anti-proliferative effect for EvA treatment.

## 3. Materials and Methods

### 3.1. Bacterial Strains

The strain *C. jejuni* ATCC 33291 , was used. The strain *C. jejuni* 11168H cdtA mutant (Table 1), obtained from the London School of Hygiene Tropical Medicine (LSHTM) *Campylobacter* Resource Facility, was added as negative control strain.

### 3.2. C. jejuni Cell Lysate Preparation

*C. jejuni* ATCC 33,291 and *C. jejuni* 11168H *cdtA* mutant strains were grown in 50 mL Brucella broth (Oxoid) at 37 °C in a shaking incubator under microaerophilic conditions for 48 h. The bacterial suspensions were adjusted spectrophotometrically to approximately 10^8^ bacteria/mL and centrifuged at 4000 rpm for 10 min. The pellets were resuspended in 20 mL of Dulbecco’s modified Eagle medium (D-MEM) (Sigma-Aldrich, St. Louis, MO, USA) and lysed by sonication (2 × 30 s bursts with 30 s intervals between each burst) by using a sonicator (Sonifier 450, Branson, MO, USA). Cell debris and unlysed bacterial cells were then removed by centrifugating at 2916 g for 10 min. Aliquots of each lysate were sterilised by a 0.22 µm membrane filter (Millipore, Burlington, MA, USA) and stored at −20 °C before use.

### 3.3. Cell Culture

U937 (human myelomonocytic cell line) cells (Sigma-Aldrich, St. Louis, MO, USA), were grown in RPMI 1640 supplemented with 10% heat-inactivated foetal bovine serum (FBS), 2 mM glutamine, and 1% antibiotics, and were maintained at 37 °C in humidified air with 5% CO_2_. Caco-2 cells were maintained at 37 °C in humidified air with 5% CO_2_ in DMEM/F12 containing 10% heat-inactivated foetal bovine serum (FBS), 2 mM glutamine, and 1% antibiotics.

### 3.4. Co-Culture

Caco-2 cells were cultured in DMEM-F12 cell culture medium supplemented with 10% heat-inactivated FBS, 1% L-glutamine, and 1% penicillin/streptomycin at 37 °C, 5% CO_2_. The U937 cells were cultured in RPMI 1640 substituted with 10% heat-inactivated FBS, 1% penicillin/streptomycin, 1% L-glutamine at 37 °C, 5% CO_2_. For co-cultures, caco-2 was seeded (2.5 × 10^5^ cells) on the apical side of the Transwell insert membrane (0.4-µm pore-size membrane) for 9 days and allowed to form polarised cell monolayers [97]. Media were changed every two days. At day 9, the Caco-2 cells were treated with *C. jejuni* lysates (*C. jejuni* ATCC 33291, and *C. jejuni* 11168H *cdtA* mutant strains) for 24 h. At day 10, the U937 cells were seeded (10^6^) in the lower chambers (scheme of co-culture at Appendix A).

### 3.5. Pre-Treatment of Cells with C. jejuni Lysates

U937 and Caco-2 cells were incubated for 24, 48, and 72 h with 2 mL of media enriched with *C. jejuni* lysates (1:100 and 1:50 dilution, respectively) from the ATCC 33,291 and 11168H *cdtA* mutant strains previously prepared. Treated cells were analysed by means of FM and CM to evaluate different cellular parameters. For the negative control, cells were incubated with media only.

### 3.6. Flow Cytometry (FC) and Confocal Microscopy (CM) Staining Assessment of Lysosomal Involvement and Lysosomal Exocytosis

To label and trace lysosomes, the acidotropic dye LysoTracker Deep Red (LTDR) (Thermo Fisher Scientific, USA) and the pH-sensitive dye Acridine Orange (AO; Sigma-Aldrich, St. Louis, MO, USA) were used at 75 ng/mL for 30 min at 37 °C in 5% CO_2_. LysoTracker probes are fluorescent acidotropic probes for labelling and tracking acidic organelles in live cells, signifying that the amount of fluorescence obtained from staining with LysoTracker is directly related to the volume of lysosome-related organelles in a cell [98]. Cells were cultured at 37 °C and resuspended in pre-warmed (37 °C) medium containing 100 nM of Lyso Tracker. After 45 min of incubation, red lysosomal fluorescence was detected by flow cytometry and confocal microscopy [99]. AO is a cell-permeable fluorescent dye that at its highest concentration stains the double stranded DNA red green and the cytoplasm green red. The dye can also enter acidic compartments such as lysosomes and autolysosomes, where it becomes protonated and sequestered. In these compartments, at its lowest concentrations, in an acid environment, AO emits red fluorescence with an intensity proportional to the degree of acidity and/or to the acidic compartment volume [15,100]. Red lysosomal and green cytoplasmic fluorescence were acquired by flow cytometry using the FL3 and FL1 channels, respectively. Cell surface CD107a (LAMP-1), which is found on lysosomes and intracellular lytic granules, was measured. LAMP-1 surface expression was used as a marker of lysosomal exocytosis [101]. The CD107a-PeCy5 antibody (clone H4A3, BioLegend, San Diego, CA, USA) was added to 50 µL of cellular suspension at the concentration indicated in the manufacturer’s instructions. Cells were incubated for 1 h at RT and analysed by flow cytometry.

### 3.7. Autophagic-like Vacuole Detection

Autophagic-like vacuoles were detected using flow cytometry by monodansylcadaverine (MDC, Sigma-Aldrich, St. Louis, MO, USA) staining, a specific autophagolysosome marker typically used to investigate the autophagic machinery [102]. Cells were incubated with 50 µM MDC at 37 °C for 15 min before proceeding to the measurements.

### 3.8. Determination of Mitochondria Membrane Potential (MMP) and Mitochondrial Reactive Oxygen Species (ROS)

Mitochondrial features were investigated by tetramethylrhodamine ethyl ester perchlorate (TMRE) (Sigma-Aldrich, St. Louis, MO, USA) and MitoSOX Red (Thermo Fisher Scientific, Waltham, MA, USA). TMRE is a MMP-specific stain able to selectively enter the mitochondria depending on MMP, producing a red fluorescence. TMRE 40 nM was added to the samples 15 min before the acquisition time. MitoSOX Red is a fluorogenic dye specifically targeted to mitochondria in live cells. Oxidation of this probe by the superoxide that is contained in the mitochondria produces a red fluorescence. A total of 5 µM of MitoSOX Red was added to the samples 10 min before the time of acquisition. The samples were analysed by confocal microscopy and flow cytometry using the appropriate fluorescence channel [9,16].

### 3.9. Evaluation of Intracellular ROS Content (Mainly H_2_O_2_)

Intracellular ROS content was measured with 5 µM 5-(and-6)-chloromethyl-2,7′-dichlorodihydrofluorescein diacetate acetyl ester (CM-H2DCFDA, DCFDA) (Molecular Probes, Eugene, OR, USA) incubated for 30 min at 37 °C [103]. Analyses were then performed by flow cytometry and confocal microscopy using the appropriate fluorescence channel.

### 3.10. Cell Cycle Analysis

The distribution of DNA in the cell cycle was investigated by flow cytometry. Cells were fixed by ice-cold ethanol (70%) and stored at 4 °C until analysis. For cell cycle analyses, samples were washed twice with PBS 1X and each pellet was resuspended in 440 µL of PBS 1X, to which 10 µL of 1 mg/mL PI (Sigma-Aldrich, St. Louis, MO, USA) and 50 µL of 1 mg/mL RNase (Sigma-Aldrich, St. Louis, MO, USA) were added to a final volume of 500 µL. Samples were resuspended and incubated at 37 °C for at least 1 h by flow cytometry.

### 3.11. Intracellular Detection of Prb, NF-kB, and PCNA

The intracellular content of molecules was revealed by previous cell fixation and permeabilisation (FIX & PERM^®^ Cell Fixation and Cell Permeabilisation Kit; Invitrogen, Thermo Fisher Scientific, Waltham, MA, USA). Cells were fixed and permeabilised, then the fixed cells were stained with monoclonal anti-human antibodies anti-pRb PE-conjugated (clone G3-245) (BD Bioscience, Franklin Lakes, NJ, USA) and RelA/NFkB p65 Alexa Fluor 488-conjugated (112A1021, Novus Biologicals, Littleton, CO, USA) was added to samples at concentrations according to the manufacturer’s instructions. Proliferating cell nuclear antigen expression was studied in cells fixed in 70% ethanol using anti-PCNA FITC-conjugated (clone P10, Sigma-Aldrich, St. Louis, MO, USA).

### 3.12. Extracellular Vesicle FACS Detection

To detect microparticles in the extracellular environment without any preparation step of ultracentrifugation, we resuspended each sample (40 µL) in 0.22-filtered PBS 1X. EVs from the *C. jejuni* lysate treated Caco-2 and co-culture medium were evaluated by means of LCD dye (BD Biosciences, Franklin Lakes, NJ, USA, Custom kit. Cat. 626267), following the flow cytometric protocols in both monoculture and in co-culture conditions [104,105]. After an incubation of 60 min to ensure the binding of the monoclonal antibodies (anti-CD107a PerCP/Cy5.5-conjugated (clone H4A3, BioLegend, San Diego, CA, USA), anti-CD63 FITC-conjugated (clone TEA3/18, Immunostep, Salamanca, Spain), and anti-CD133 PE-conjugated (clone 293C3, Miltenyi Biotec Bergisch Gladbach, Germany) to the specific epitope, we proceeded with FC analyses. The flow cytometry approach consisted of acquiring samples mixed with beads of defined size (Ø 0.5 µm, 1 µm, 5.2 µm) to obtain the size calibration of small particles detected outside the scatter area of intact cells. Furthermore, it is important to specify that the FCS of the complete medium was ultracentrifuged to minimise contamination by serum microvesicles. Samples were acquired by a FACSCanto II flow cytometer and FACSDiva™ software was used to analyse all data.

### 3.13. Scanning Electron Microscopy

Caco-2 cells from the Transwell insert membrane were cut from the supports and fixed with 2.5% glutaraldehyde in 0.1 M phosphate buffer pH 7.2–7.4 for 1 h, postfixed with 1% osmium tetroxide (OsO4) in the same buffer for 2 h, and finally dehydrated with graded ethanol (50%–100%, 5 min each). Critical point dried specimens were mounted on aluminium stubs. After 10 nm, gold sputter-coated samples were examined with a Philips Scanning Electron Microscope (SEM) at 20 kV [106].

### 3.14. Extracellular Vesicle Isolation

Conditioned media were collected from the Caco-2 cell cultures after 72 h of lysate treatment in EV-depleted media. Conditioned media were only used for EV collection. EV isolation was performed following the guidelines reported in Turchinovich et al. (2012) [107]. Briefly, the medium was first cleared by centrifugation for 15 min at 1000× *g* to eliminate cell contamination. Supernatants were further centrifuged for 20 min at 12,000× *g* and subsequently for 20 min at 18,000–20,000 g. The resulting supernatants were filtered through a 0.22 µm filter and then the EVs were pelleted by ultracentrifugation at 110,000× *g* for 70 min. The EV pellets were washed in PBS, pelleted again, and resuspended in PBS.

### 3.15. Nanoparticle Tracking Analysis (NTA)

The EV preparations were characterised according to the guidelines of the International Society for Extracellular Vesicles [108]. The EV size and concentration were measured by nanoparticle tracking analysis. Isolated EVs were diluted at 1:100 in PBS and stored at −20 °C for further analysis. NTA measurements were performed with a NanoSight LM20 (NanoSight, Malvern Instruments Ltd., Malvern, UK), equipped with a sample chamber with a 640 nm laser and a Viton fluoroelastomer O-ring. The samples were injected in the sample chamber with sterile syringes until the liquid reached the tip of the nozzle. All measurements were performed at room temperature.

### 3.16. miRNA Quantification

The EV pellets were washed, pelleted again, and resuspended in PBS for the nanoparticle tracking assay before lysis with Trizol for RNA extraction. The extraction of miRNA from the EV pellets was performed using the Total Exosome RNA and Protein Isolation Kit (Invitrogen). Reverse transcription of RNA was performed using the miRCURY LNA Universal RT microRNA PCR, Polyadenylation, and cDNA Synthesis Kit II, according to the manufacturer’s instructions (Exiqon, Milan, Italy). Real-time quantitative PCR for the cDNA extracted from the EVs was performed in a StepOnePlus Real-Time PCR (Applied Biosystems, Italy) using the ExiLENT SYBR^®^ Green master mix and the miRCURY LNA miRNA PCR assays (Exiqon, Vedbaek, Denmark), specific for human miR-146a-5p and miR-16-5p [109].

### 3.17. Extracellular Vesicle PKH67 Staining

EVs released from Caco-2 treated by lysates were fluorescently labelled using PKH67 (Green Fluorescent Cell linker for General Cell Membrane) according to the manufacturer’s instructions (Sigma-Aldrich, St. Louis, MO, USA). Briefly, PKH-67 green fluorescence was added at 2–4 µM EV suspension in diluent C and the mixture was incubated at room temperature for 5 min. Staining was stopped using 1% BSA and the EV suspension was subjected to ultracentrifugation at 100,000× *g* for 2 h.

### 3.18. Analysis of the In Vitro EV Uptake

To evaluate the uptake of EVs, U937 and Caco-2 cells were incubated for 24, 48, and 72 h with 1 mL of media enriched with 2 × 10^4^ PKH67-labeled EVs/cell isolated from a medium of Caco-2 cells treated with different lysates: ATCC 33,291 and the 11168H *cdtA* mutant [110,111]. For the negative control, cells were incubated with media only. EV treated cells were analysed by means of flow cytometry to evaluate different cellular parameters and confocal microscopy to assess in vitro colocalisation studies.

### 3.19. Flow Cytometric Absolute Count

Absolute cell counting was performed by using Dako CytoCountTM beads (Thermo Fisher Scientific, Waltham, MA, USA). A total of 200 µL of the sample was carefully dispensed at the bottom of the tube and 50 µL beads were added. Samples were acquired by using a FACSCanto II cytometer (BD, USA) within 60 min. Approximately 20,000 cell events were collected. Setup and calibration procedures were optimised for the absolute counting protocols [112]. Additionally, absolute counting was performed by an Omnicyt flow cytometer (Cytognos SL, Salamanca, Spain).

### 3.20. Cytometric Investigations

Cytometric experiments were carried out with a FACSCanto II flow cytometer (BD, Franklin Lakes, NJ, USA) equipped with an argon laser (Blue, Excitation 488 nm), a helium-neon laser (Red, Excitation 633 nm), a solid-state diode laser (Violet, Excitation 405 nm), and Omnicyt flow cytometer (Cytognos SL, Salamanca, Spain). Analyses were performed with the FACSDivaTM software (BD) and the Infinicyt 2.0 software (Cytognos SL). Approximately 10,000 cell events were acquired for each sample.

### 3.21. Confocal Microscopy

Confocal microscopy analyses were applied by a Leica TCS SP5 II confocal microscope (Leica Microsystem, Germany) with 488, 543, and 633 nm illuminations and oil-immersed objectives. For confocal live imaging, cells were grown on MatTek glass bottom chambers (MatTek Corporation, Bratislava, Slovak Repu). The images were further processed and analysed in ImageJ software (NIH, Bethesda, MD, USA).

### 3.22. Statistical Analyses

Data are shown as the mean (or percentage, as indicated) ± standard deviation (SD) of at least three independent experiments. Analysis of variance (ANOVA) approaches were used to compare values among more than two different experimental groups for data that met the normality assumption. One-way ANOVA or two-way ANOVA were followed by a Bonferroni post hoc test. The p values less than 0.05 were considered statistically significant. Bonferroni’s multiple comparison test revealed statistical significance: * = *p* < 0.05, ** = *p* < 0.01, *** = *p* < 0.001. All statistical analyses were performed using GraphPad Prism 9.0.0 (GraphPad software, San Diego, CA, USA).

## 4. Conclusions

We demonstrate here that the EV cargo contains active CDT, which is able to affect various recipient cells differently. Proliferation is distinctly reduced in EV administered Caco-2 and U937 cells, although with a different extent in the two cell lines. Homologous EVs (released by Caco-2 cells and internalised by the same intestinal cell line) appear to be the most efficient system for both uptake and antiproliferative response.

In agreement with other researchers [113], we demonstrated that bacterial toxins are suitable tools in cancer therapy, and highlighted both the benefits and limitations of this application. Global cell response involves a moderate induction of apoptosis and autophagic features, which may play a protective role against toxin-induced cell death [16]. Moreover EVs represent affordable CDT carriers, which are particularly suitable in colorectal adenocarcinoma Caco-2 cells. In fact, EVs are not only pertinent biological markers of CRC, but are also potential targets to inhibit cancer [114].

Finally, the unexpected cell cycle behaviour of EVs from the *C. jejuni* mutant CDT treated Caco-2 cells highlights one limitation of the study: the toxin derives from a mutated strain for cdtA but not for cdtB. In addition, we did not utilise the purified form of the toxin. However, our data propose a system with a potential novel bacterial-related biotherapeutic that is applicable in supporting multidrug anticancer protocols.

## Figures and Tables

**Figure 1 ijms-24-00487-f001:**
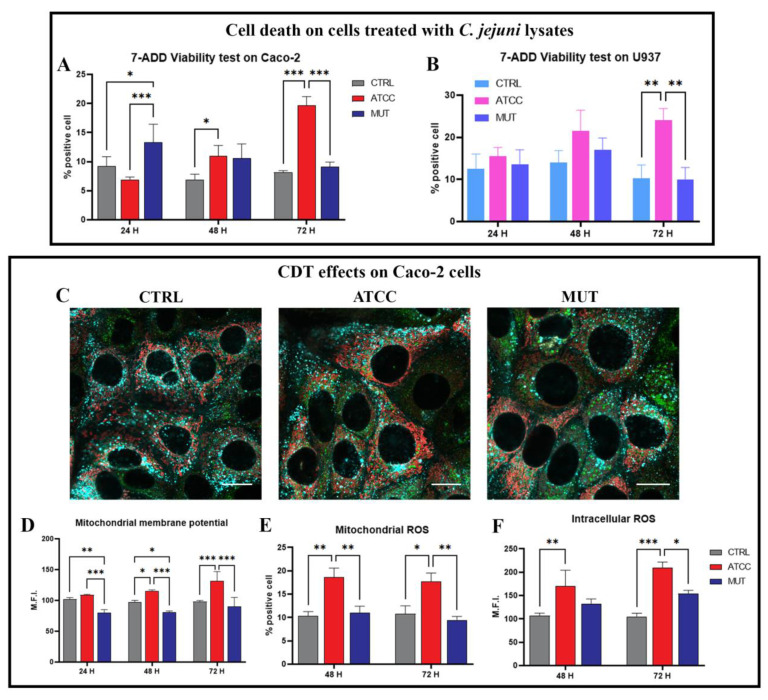
Evaluation of cell death, mitochondria dysfunction, and ROS production. (**A**) Statistical histograms of the percentage of 7′AAD positive cells from 24 h to 72 h on Caco-2 cells treated with *C. jejuni* lysates. (**B**) Statistical histograms of the percentage of 7′AAD positive cells from 24 h to 72 h on U937 cells treated with *C. jejuni* lysates. (**C**) Single confocal optical sections displaying an overlay of mitochondria (red), lysosome (cyan), and lipid droplets (green) at 48 h. Bars: 20 µm. (**D**) Statistical histograms of the TMRE MFI value from 24 h to 72 h lysate administration on Caco-2 cells. (**E**) Statistical histograms of the MitoSOX MFI value from 48 h to 72 h lysate administration on Caco-2 cells. (**F**) Statistical histograms of CM-H2DCFDA MFI value from 48 h to 72 h lysate administration on Caco-2 cells. Each value is expressed as a mean ± SD (results from n ≥ three independent experiments). Two-way ANOVA with Bonferroni’s multiple comparison test revealed significant difference * = *p* < 0.05, ** = *p* < 0.01, *** = *p* < 0.001.

**Figure 2 ijms-24-00487-f002:**
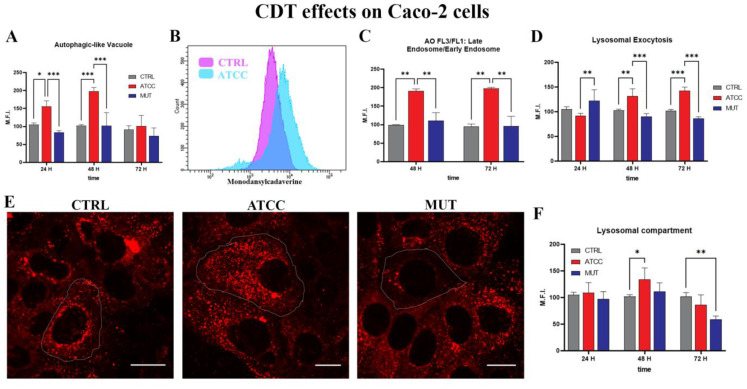
Evaluation of autophagic-like and endo-lysosomal involvement. (**A**) Statistical histogram of MDC MFI from 24 h to 72 h. (**B**) Representative cytometric histogram of the MDC channel for CTRL in violet and ATCC in cyan. (**C**) Statistical histograms of the ratio of the FL3/FL1 AO value at 48 h and 72 h after lysate administration. (**D**) Statistical histograms of the LAMP-1 MFI value at 24 h, 48 h, and 72 h after lysate administration. (**E**) Single confocal optical sections show the LTDR (red) for lysosome tracking at 48 h for the untreated-control cells (CTRL), cells treated by ATCC *C. jejuni* lysates (ATCC), and mutant *cdtA C. jejuni* (MUT). Bars: 20 µm. (**F**) Statistical histograms of the LTDR MFI value from 24 h to 72 h of the lysate treated Caco-2 cells. Two-way ANOVA with Bonferroni’s multiple comparison test. The asterisk denotes a statistically significant difference (* = *p* < 0.05, ** = *p* < 0.01, *** = *p* < 0.001).

**Figure 3 ijms-24-00487-f003:**
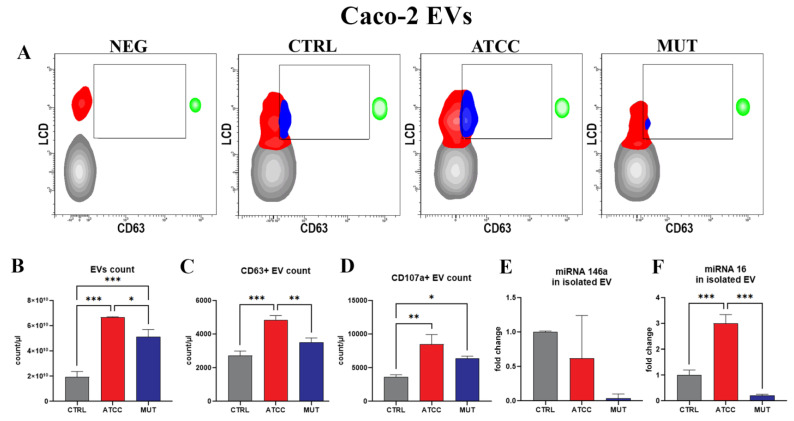
(**A**) Density plot of EVs from the Caco-2 cells of NEG, CTRL, ATCC, and MUT at 72 h. Red area indicates EV positive for LCD, blue area indicates EV positive for CD63. Green area identifies cytometer counting beads. (**B**) Histogram of EV count at 72 h. (**C**) Histogram of CD63 positive EV at 72 h. (**D**) Histogram of CD107a+ EV at 72 h. (**E**) Histogram of EV-miRNA-146a content. (**F**) Histogram of EV-miRNA-16 content. One-way ANOVA with Bonferroni’s multiple comparison test. The asterisk denotes a statistically significant difference (* = *p* < 0.05, ** = *p* < 0.01, *** = *p* < 0.001).

**Figure 4 ijms-24-00487-f004:**
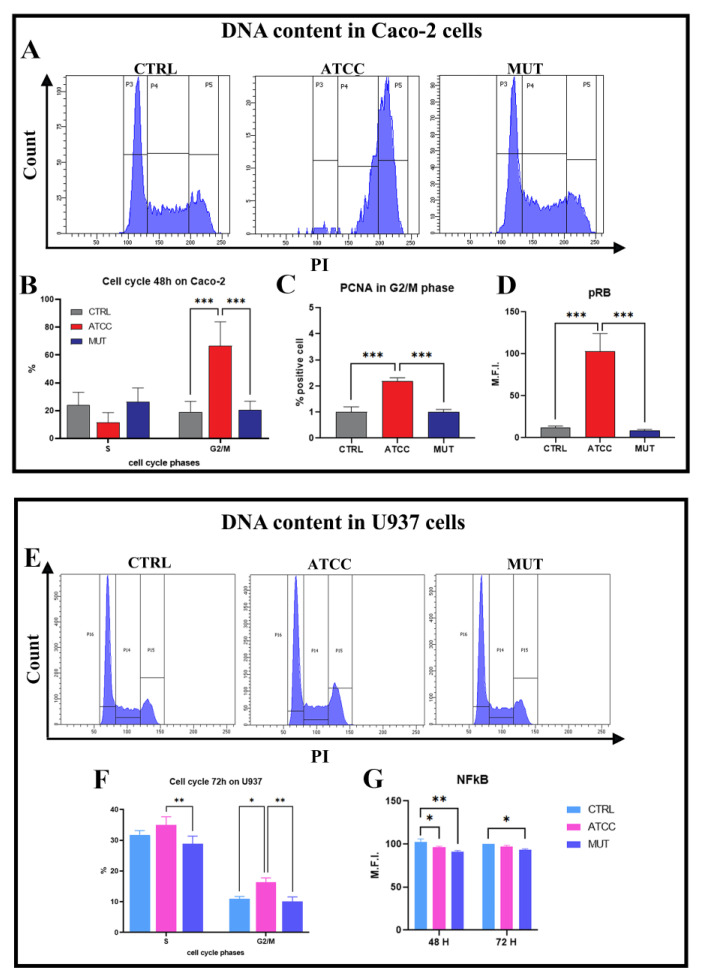
Evaluation of the efficiency of the toxin and induction of the cell cycle block. (**A**) FC histograms represent the Caco-2 cell population in the G0/G1 (P3), S (P4), and G2/M (P5) phases of cell cycle after 48 h of lysate administration. (**B**) Statistical histogram of the cell cycle phases calculated in cytometry via PI staining at 48 h. (**C**) Statistical histograms of the percentage of PCNA positive cells at G2/M phase cells at 72 h. (**D**) Statistical histograms of intracellular pRB MFI cells at 72 h. (**E**) FC histograms represent the U937 cell population in the G0/G1 (P16), S (P14), and G2/M (P15) phases of the cell cycle after 72 h of lysate administration. (**F**) Statistical histogram of the S and G2/M phases of the cell cycle calculated in cytometry via PI staining at 72 h. (**G**) Statistical histograms of NF-kB p65 MFI cells at 48 and 72 h. One or two-way ANOVA with Bonferroni’s multiple comparison test. The asterisk denotes a statistically significant difference (* = *p* < 0.05, ** = *p* < 0.01, *** = *p* < 0.001).

**Figure 5 ijms-24-00487-f005:**
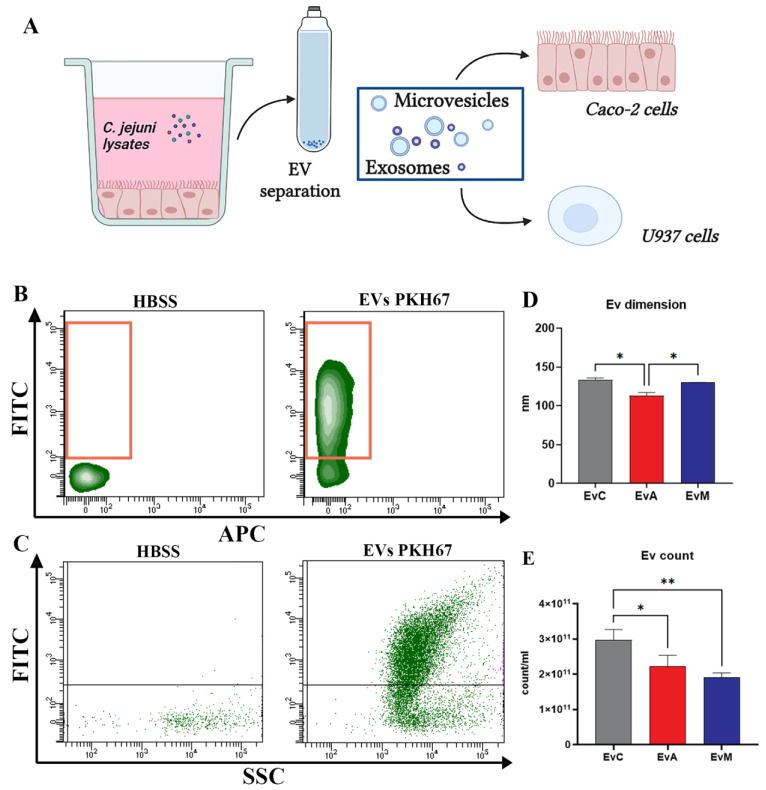
(**A**) Schematic of EV separation by ultracentrifugation and EV administration to the uninfected Caco-2 and U937 cells (created in Biorender.com). (**B**) Density plot FITC vs. APC of the HBSS and PKH67-labeled EVs. (**C**) Dot plot FITC vs. SSC of the HBSS and PKH67-labelled EVs. (**D**) Statistical histogram for the size distribution for EvC, EvA, and EvM. (**E**) Statistical histogram for the particle concentration for EvC, EvA, and EvM. One-way ANOVA with Bonferroni’s multiple comparison test. The asterisk denotes a statistically significant difference (* = *p* < 0.05, ** = *p* < 0.01).

**Figure 6 ijms-24-00487-f006:**
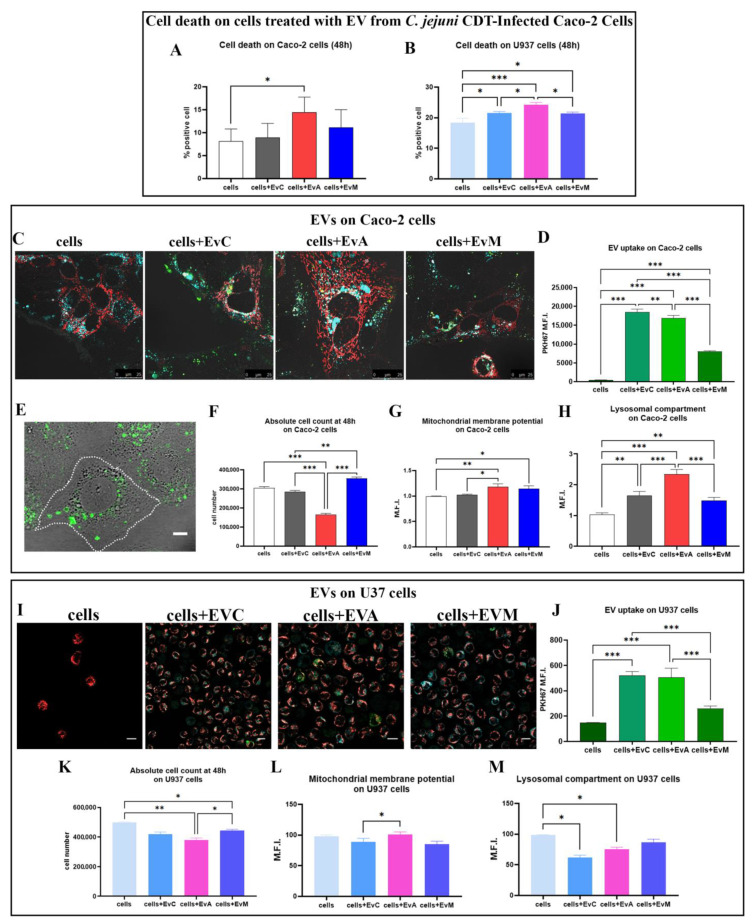
(**A**) Statistical histograms of the percentage of cell death on Caco-2 cells treated with EVs from the *C. jejuni* CDT-infected Caco-2 cells after 48 h. (**B**) Statistical histograms of the percentage of cell death on the U937 cells treated with EVs from the *C. jejuni* CDT-infected Caco-2 cells after 48 h. (**C**) Single confocal optical sections display an overlay of the mitochondria (TMRE red), lysosome (LTDR cyan), and EVs (PKH67 green) at 48 h for the cells, cells + EvC, cells + EvA, and cells + EvM. (**D**) Statistical histogram of the uptake of PKH67-labeled EV for the Caco-2 cells treated with EvC, EvA, and EvM at 48 h. (**E**) Representative micrograph of the EV-labelled PKH-67 (green) internalisation on Caco-2 cells. Bar: 20 µm. (**F**) Statistical histogram of the cell count at 48 h. (**F**) Statistical histogram of the cell count at 48 h on the Caco-2 cells. (**G**) Statistical histograms of TMRE MFI value on cells, cells + EvC, cells + EvA, and cells + EvM. (**H**) Statistical histograms of the LTDR MFI value for cells, cells + EvC, cells + EvA, and cells + EvM. (**I**) Single confocal optical sections show an overlay of the mitochondria (TMRE red), lysosome (LTDR cyan), and EV (PKH67 green) at 48 h for the U937 cells, cells + EvC, cells + EvA, and cells + EvM. (**J**) Statistical histogram of PKH67-labelled EV uptake for Caco-2 cells treated by EvC, EvA, and EvM at 24 and 48 h. (**K**) Statistical histogram of the cell count at 48 h on the U937 cells. (**L**) Statistical histograms of the TMRE MFI value on the cells, cells + EvC, cells + EvA, and cells + EvM at 24 h and 48 h. (**M**) Statistical histograms of the LTDR MFI value for the cells, cells + EvC, cells+ EvA, and cells + EvM. One-way ANOVA with Bonferroni’s multiple comparison test. The asterisk denotes a statistically significant difference (* = *p* < 0.05, ** = *p* < 0.01, *** = *p* < 0.001).

**Figure 7 ijms-24-00487-f007:**
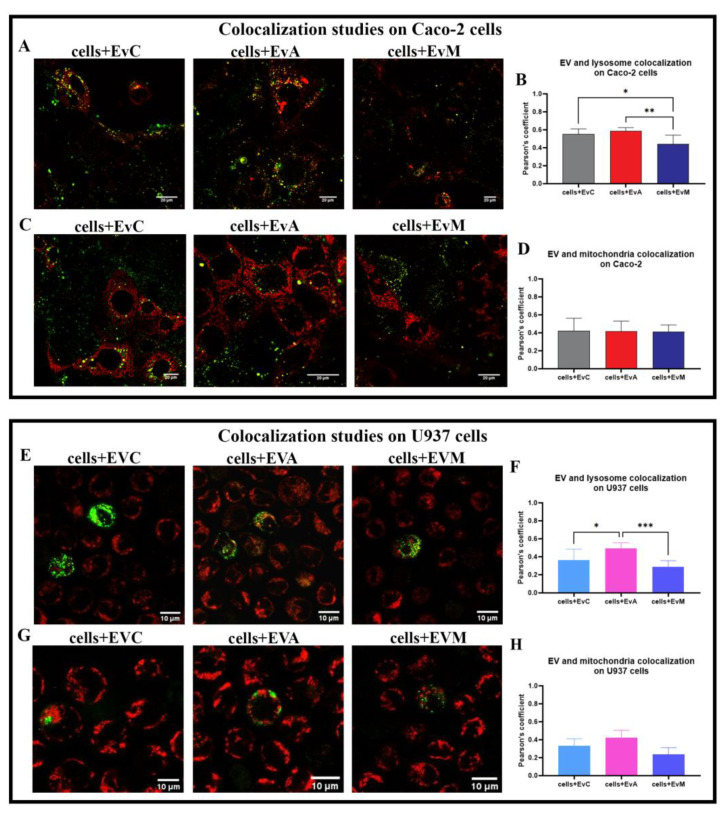
(**A**) Single confocal optical sections of Caco-2 cells showing an overlay of the lysosome (LTDR red) and EVs (PKH67 green) at 48 h for the cells + EvC, cells + EvA, and cells + EvM. Colocalisation of the EVs (green) and lysosomes (red) is displayed in yellow. The bar indicates 20 µm. (**B**) Pearson’s coefficient, able to quantitate LTDR/EV co-localisation for the cells + EvC, cells + EvA, and cells + EvM. Pearson’s coefficients were derived from three biological replicates with >10 fields per experiment contributing to the cumulative result. (**C**) Single confocal optical sections of Caco-2 showing an overlay of the mitochondria (TMRE red) and EVs (PKH67 green) at 48 h for the cells + EvC, cells + EvA, and cells + EvM. The bar indicates 20 µm. (**D**) Pearson’s coefficient, able to quantify TMRE/EV co-localisation for the cells + EvC, cells + EvA, and cells + EvM. Pearson’s coefficients were derived from three biological replicates with >10 fields per experiment contributing to the cumulative result. (**E**) Single confocal optical section of U937 displaying an overlay of the lysosomes (LTDR red) and EVs (PKH67 green) at 48 h for the cells + EvC, cells + EvA, and cells + EvM. Colocalisation of the EVs (green) and lysosomes (red) is displayed in yellow. The bar indicates 10 µm. (**F**) Pearson’s coefficient, able to quantitate LTDR/EV co-localisations for the cells + EvC, cells + EvA, and cells + EvM. Pearson’s coefficients were derived from three biological replicates with >10 fields per experiment contributing to the cumulative result. (**G**) Single confocal optical section of U937 displaying an overlay of the mitochondria (TMRE red) and EVs (PKH67 green) at 48 h for the cells + EvC, cells + EvA, and cells + EvM. Colocalisations of the EVs (green) and mitochondria (red) are displayed in yellow. The bar indicates 10 µm. (**H**) Pearson’s coefficient, able to quantitate TMRE/EV co-localisations for the cells + EvC, cells + EvA, and cells + EvM. Pearson’s coefficients were derived from three biological replicates with >10 fields per experiment contributing to the cumulative result. One-way ANOVA with Bonferroni’s multiple comparison test. The asterisk denotes a statistically significant difference (* = *p* < 0.05, ** = *p* < 0.01, *** = *p* < 0.001).

**Figure 8 ijms-24-00487-f008:**
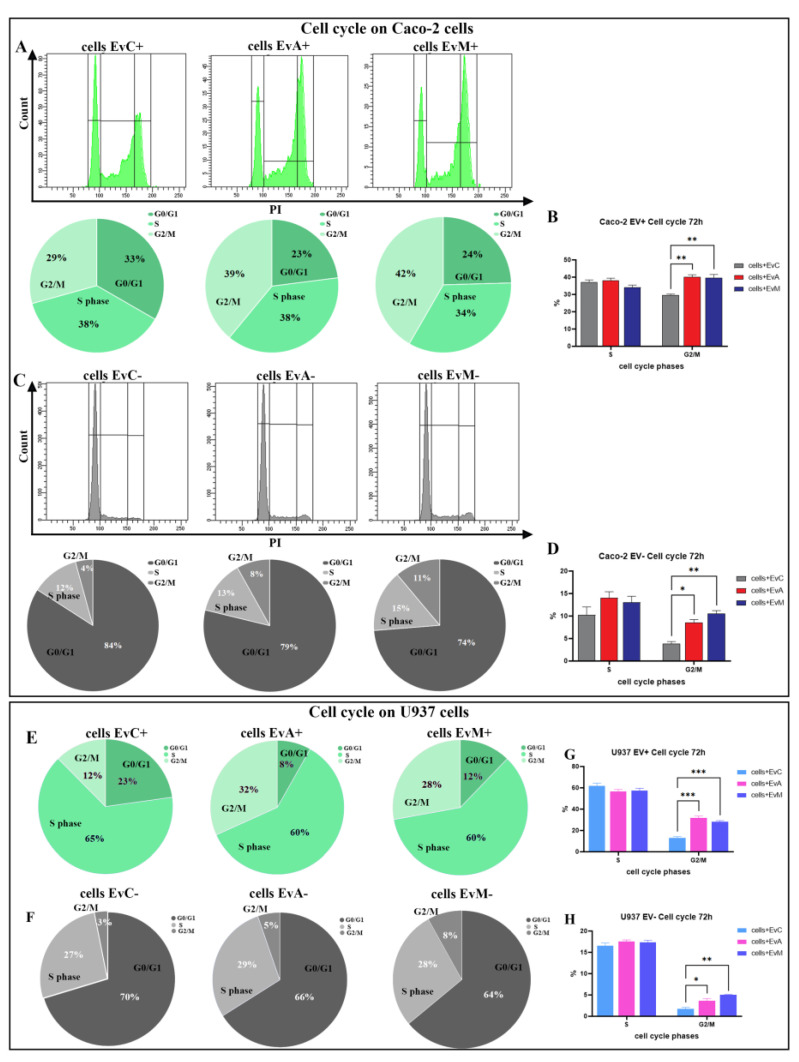
(**A**) Cytometric histograms (above) and pie chart (below) for the cell cycle distribution of Caco-2 cells with detected EvC+, cells with detected EvA+, cells with detected EvM+. (**B**) Statistical histogram of the S and G2/M phases of cell cycle calculated in cytometry via PI staining at 72 h in Caco-2 cells with detected EV+. (**C**) Cytometric histograms (above) and pie chart (below) for the cell cycle distribution of Caco-2 cells with undetected EvC, cells with undetected EvA, cells with undetected EvM. (**D**) Statistical histogram of the S and G2/M phases of cell cycle calculated by flow cytometry via PI staining at 72 h in Caco-2 cells with undetected PKH67 EV (EV−). (**E**) Pie chart of the cell cycle distribution on U937 cells with detected EV+. (**F**) Statistical histogram of the S and G2/M phases of the cell cycle of cells with detected EV calculated in cytometry via PI staining at 72 h. (**G**) Pie chart of the cell cycle distribution on U937 cells with undetected EV−. (**H**) Statistical histogram of the S and G2/M phases of the cell cycle of cells with undetected EV− calculated in cytometry via PI staining at 72 h. Two-way ANOVA with Bonferroni’s multiple comparison test. The asterisk denotes a statistically significant difference (* = *p* < 0.05, ** = *p* < 0.01, *** = *p* < 0.001).

**Table 1 ijms-24-00487-t001:** Bacterial strain description and references.

*C. jejuni* Strains	Description	References
11168H *cdtA* mutant	Isogenic mutant of *Cj0079* obtained from the Campylobacter mutant bank (LSHTM)	LSHTM mutant bank
ATCC 33291	*Campylobacter jejuni* subsp. *jejuni* (ATCC® 33291™)	ATCC^®^, Rockville, ND, USA

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
