# Peer review of "Extracellular Vesicles from Campylobacter jejuni CDT-Treated Caco-2 Cells Inhibit Proliferation of Tumour Intestinal Caco-2 Cells and Myeloid U937 Cells: Detailing the Global Cell Response for Potential Application in Anti-Tumour Strategies"

_ijms, 2022, doi:10.3390/ijms24010487_

Round 1

Reviewer 1 Report

Regarding the manuscript “Extracellular vesicles from Campylobacter jejuni CDT intoxi-2 cated-Caco-2 cells for potential application in antitumor strate-3 gies: cargo delivery inhibits proliferation in tumor intestinal and 4 myeloid cells”

            I must congratulate the authors for a wonderful idea. The natural compounds like EVs that can be used as drug carriers are not cutting edge but for sure novel. Nevertheless, the use EVs from bacteria to use on antitumor treatment is indeed cutting edge.

            The manuscript is full of various techniques that were used. There are a lot of pictures, schemes, and graphs that allow the reader to know the topic as well as the proof of the presented concept itself. However, there is still a need to make some updates in order for this paper to get published.

            Major:

            The presentation style must be updated. It is missing the “sharpness” of the idea. When reading the manuscript you have to search for the main idea in order to understand what the authors wanted to present. The reviewer strongly recommends the authors make the research story in the manuscript clear. It was a real struggle to understand the main results and ideas.

            The reviewer did not find an explanation of ATCC, MUT, CTRL (probable control), or NEG. Only after reading the almost full manuscript, I understood that it is coding for C.jejuni. If there is a shortening please explain it before it is used. There is no section about the bacteria at all – only a lysate preparation protocol.

            What do the colors in fig.2 A mean? It is confusing that the same colors are used in fig 2 part B-F but the meaning probably is different than in part A.

            The markings of statistical significance are improper. If the author prefers to mark it this way please explain the markings in the captions and not only in the methodology.

            Why miRNA 16 in isolated EV work on ATCC but not on MUT in fig. 2 F?

            The scheme in fig. 2 G is unreadable. Please enlarge it.

Why U937 cells were started to be used only from fig 3?

Fig 3 is done unproperly. Y axis in A and E parts are not in the correct 90 degree angle I can read A part x axis but E part axis is cut in half.

Why Caco-2 is measured after 48 hours but U937 after 72 hours? As it is cell cycle measurements it is crucial to have good timing.

Are colors in fig 3 also random? If so rearrange it because it is confusing.

Fig 4 why SEM monographs are only Caco-2? U937 also should be included.

What is the difference between fig 3 F and fig 4 D-F? The results differ but the description of what was done is practically the same.

Fig 5 A is not in good resolution. Please change it. Fig 5 B-D axis values are practically unreadable. Please enlarge it.

How did the authors separate the EVs into EVc EvA and EVm? It is crucial since other results depend on it.

What is the difference between figure 8 part A and C ? In the captions, it says it is the same. If it is the same way such a huge difference?

In fig 9 apoptosis is mentioned. I did not find the graphs for it. Did the authors do something with apoptosis measurements?

Regarding cell death. NO CELL VIABILITY assays were done or I did not find it. That is the MAIN thing in this publication. Anticancer treatment means cancer cell death.

The same goes for DNA damage mentioned in the conclusions. Have the authors done it?

  Please present the ROS graph in the manuscript since you have put all sections on this topic it must be in the main text of the manuscript.

All the manuscript is chaos one must have a lot of stamina to read it through and try to find the main ideas the authors probably wanted to express. It really needs rewriting.

Author Response

            Major:

  1. The presentation style must be updated. It is missing the “sharpness” of the idea. When reading the manuscript, you have to search for the main idea in order to understand what the authors wanted to present. The reviewer strongly recommends the authors make the research story in the manuscript clear. It was a real struggle to understand the main results and ideas.

We thank the reviewer for their major comment. We have followed the request and have rewritten the manuscript with more clarity and have presented our results and ideas in a much more succinct format. We have also streamlined our manuscript reducing the number figures. All changes are evident in our revamped manuscript. Finally, we have also adjusted the title, to reflect the convey the context of our study more punctually to the reader. 

  1. The reviewer did not find an explanation of ATCC, MUT, CTRL (probable control), or NEG. Only after reading the almost full manuscript, I understood that it is coding for jejuni. If there is a shortening, please explain it before it is used. There is no section about the bacteria at all – only a lysate preparation protocol.

We have added the explanation of the bacteria strains as requested.

  1. What do the colors in fig.2 A mean? It is confusing that the same colors are used in fig 2 part B-F but the meaning probably is different than in part A.

Thank you for this observation. The significance if different and has now been explained in the caption. Furthermore, coloured figures and histograms are now reduced in number, leaving only those necessary for our message.

  1. The markings of statistical significance are improper. If the author prefers to mark it this way please explain the markings in the captions and not only in the methodology.

We adjusted the mentioning of statistical significance, and we believe our figures are far clearer to the reader.

  1. Why miRNA 16 in isolated EV work on ATCC but not on MUT in fig. 2 F?

We added the results of miRNA-16 in isolated EV from medium of cells treated with MUT. We had initially excluded it for the negligible amount detected.

  1. The scheme in fig. 2 G is unreadable. Please enlarge it.

We have removed the scheme which was not essential for explaining our results.

  1. Why U937 cells were started to be used only from fig 3?

Thank you for this question which allows us to refer to our previous research. In fact, we have already published a manuscript of C. jejuni lysates effects on U937 cells, specifically entitled ”Rapamycin Re-Directs Lysosome Network, Stimulates ER-Remodeling, Involving Membrane CD317 and Affecting Exocytosis, in Campylobacter Jejuni-Lysate-Infected U937 Cells” by Canonico, B., Cesarini, E., Montanari, M., Di Sario, G., Campana, R., Galluzzi, L., Sola, F., Gundogdu, O., Luchetti, F., Diotallevi, A., Baffone, W., Giordano, A., & Papa, S. (2020 International journal of molecular sciences, 21(6), 2207. Therefore, in this manuscript we have inserted only novel findings, which are functionally related to the study.

  1. Fig 3 is done unproperly. Y axis in A and E parts are not in the correct 90 degree angle I can read A part x axis but E part axis is cut in half.

Thank you for the suggestion. We have adjusted this figure.

  1. Why Caco-2 is measured after 48 hours but U937 after 72 hours? As it is cell cycle measurements it is crucial to have good timing.

We differentiated time of analysis in the two cell lines because we performed a previous study (Canonico et al., 2020) on U937 cells, underlining that relevant effects that are observed at 72 h, whereas at 48 h we start to observe several CDT effects on Caco-2 cells. We added (in this point-by-point response) an additional cell cycle profile for U937 cells, after 48 h of lysate administration.

  1. Are colors in fig 3 also random? If so rearrange it because it is confusing.

We have rearranged the figure and the panel now appears more accurate and in order.

  1. Fig 4 why SEM monographs are only Caco-2?

We have added in this point-by-point response a panel of U937 cells from the co-culture experiments. However, to simplify the overall messages from the manuscript, we have moved the panel from the main text to the supplementary material. Thus, limiting the discussion of aspects that are not functional to express the key ideas of our work.

scale bar 5µm

  1. What is the difference between fig 3 F and fig 4 D-F? The results differ but the description of what was done is practically the same.

In figure 3F, we present the cell cycle alteration of U937 cells treated with C. jejuni lysates, whereas in figure 4 D-F (now moved in the supplementary section) we have shown alterations of cell cycle phases induced by the co-culture condition and not by the direct treatment with lysates.

  1. Fig 5 A is not in good resolution. Please change it. Fig 5 B-D axis values are practically unreadable. Please enlarge it.

We have enlarged Fig 5 following your suggestion, and remove the Fig. 5 B-D.

  1. How did the authors separate the EVs into EVc EvA and EVm? It is crucial since other results depend on it.

We derived EVs into EVc EvA and EVm from different preparations by ultracentrifugation, as detailed in MM Session. In fact, we collected EVs from the medium of Caco-2 untreated (control-EVc) cells, from C. jejuni ATCC 33291 lysate treated Caco-2 (EvA), and from C. jejuni 11168H cdtA mutant lysate-treated Caco-2 cells (EvM). Then we labelled the isolated EV by PKH67, and we analysed by NTA and FC, before their administration to the cells.

  1. What is the difference between figure 8 part A and C ? In the captions, it says it is the same. If it is the same way such a huge difference?

Firstly, we have rewritten the caption. However, the goal is to distinguish two subsets of cell:

  • cells with clearly detectable PKH67 fluorescent EVs (EV+) and;
  • cells with undetectable PKH67 fluorescent EVs (EV-). These cells do not reveal PKH67 EVs for two principal reasons, i) they could not be internalised; or ii) they could be processed and/or released by cells before the time of analysis. For the latter condition, evaluating changes in cell cycle profile becomes relevant.

The two subsets were analysed at 72 h to better discriminate alterations in cell cycle phases in both Caco-2 and U937 cells.

  1. In fig 9 apoptosis is mentioned. I did not find the graphs for it. Did the authors do something with apoptosis measurements?

Thank you for your question. We had already performed cell viability tests and now we added data on cell death, for both CDT/lysate and EVs treatments.

  1. Regarding cell death. NO CELL VIABILITY assays were done or I did not find it. That is the MAIN thing in this publication. Anticancer treatment means cancer cell death.

As stated in our previous point, we have added the cell death analysis to the manuscript. However as published by previous research (Canonico et al., 2020), this toxin does not produce high percentages of cell death, particularly into 48 h treatments. Percentage cell death increases after 48 h (particularly at 72 h and 96 h).

  1. The same goes for DNA damage mentioned in the conclusions. Have the authors done it?

We have deleted the sentence from the conclusions. However, although not specifically investigating DNA damage (we will apply the COMET assay in future experiments) the cell cycle blocking at G2/M phase (a typical feature of CDT) would reduce the ability of DNA damage repair (Guo et al., 2021).

  1. Please present the ROS graph in the manuscript since you have put all sections on this topic it must be in the main text of the manuscript.

We have inserted the graph in the text.

  1. All the manuscript is chaos one must have a lot of stamina to read it through and try to find the main ideas the authors probably wanted to express. It really needs rewriting.

We followed the suggestions and rewrote the manuscript more fluently and specifically indicating the main results and ideas with more clarity. We have also reduced the amount of data and figures in order not to lose the main message of the study. Finally, we have divided the work into two parts for more clear division of our study.

References

Canonico, B., Cesarini, E., Montanari, M., Di Sario, G., Campana, R., Galluzzi, L., Sola, F., Gundogdu, O., Luchetti, F., Diotallevi, A., Baffone, W., Giordano, A., Papa, S., 2020. Rapamycin re-directs lysosome network, stimulates er-remodeling, involving membrane CD317 and affecting exocytosis, in Campylobacter Jejuni-lysate-infected U937 cells. Int. J. Mol. Sci. 21, 2207. https://doi.org/10.3390/ijms21062207

Guo, X.X., Guo, Z.H., Lu, J.S., Xie, W.S., Zhong, Q.Z., Sun, X.D., Wang, X.M., Wang, J.Y., Liu, M., Zhao, L.Y., 2021. All-purpose nanostrategy based on dose deposition enhancement, cell cycle arrest, DNA damage, and ROS production as prostate cancer ra-diosensitizer for potential clinical translation. Nanoscale 13, 14525–14537. https://doi.org/10.1039/D1NR03869A

Reviewer 2 Report

In this manuscript, the authors aimed to characterize that the extracellular vesicles from C. jejuni CDT intoxicated-Caco-2 cells for potential application in antitumor strategies. They found that either the wild-type C. jejuni lysate or extracellular vesicles from C. jejuni CDT intoxicated-Caco-2 cells induce cell cycle arrest in G2/M phase whereas this does not take place in CdtA mutant-treated cells. These findings may provide approaches to deliver therapeutic cargoes in anticancer strategies. While the findings are interesting for the field, the manuscript is desired to be revised due to some major and minor concerns as follows: 

1. For the introduction, it is suggested to divide it into different paragraphs so as to introduce it hierarchically, which will help the readers to grasp the current situation and significance of this study.

2. In lines 93-94, “This study focuses on mitochondrial modifications, deep changes in lysosomal exocytosis, secretory autophagy and EV release.....”. However, in the introduction, the authors did not mention the effects of CDT on these biological processes and its corresponding biological significance.

3. In general, the experimental design in this article is unreasonable:

--C. jejuni CDT intoxicated-Caco-2 cells should be Caco-2 cells treated with CDT from C. jejuni, not C. jejuni lysate;

--CDT toxicity is mainly dependent on CdtB activity, so why use the CdtA mutant as the control?

--Please provide the details for the strains C. jejuni ATCC 33291 and C. jejuni 11168H CdtA mutant.

-- How to select the dose and time point for C. jejuni lysate and EVs?

--For NF-κB signaling pathway activation is usually analyzed by detecting its phosphorylation status and transcriptional activity;

--For Rb protein, its effect on cell proliferation is usually analyzed by detecting its phosphorylation status using Western blot.

-- The extracellular vesicles from C. jejuni CDT intoxicated-Caco-2 cells inhibit proliferation of tumor intestinal and myeloid cells, is it tumor-specific? Does it kill the normal intestinal and myeloid cells?  

--How to demonstrate the EVs from infected Caco-2 cells contain CDT, and the following phenomenon of cell cycle arrest is caused by CDT uptake?

4. The identification evidence of extracellular vesicles is insufficient, and please provide the electron microscopic (TEM or SEM) results.

5. Keshtvarz et al. produced an engineering of CdtB to reduce immunogenicity and maintain stability as a new drug candidate for tumor therapy. Please discuss the advantages and disadvantages of this study in depth.

6. Line 117 “ATCC treated cells”, what is ATCC?

7. The horizontal and vertical coordinates of the histogram are not standardized, such as “%”.

8. Please improve your figure legends, for example, what is the meaning of ' ATCC, Mut ' in Figure 1 A? what is the meaning of the Statistical analysis Symbol marked in the figures?

9. Please improve your figures, for example, the text in the picture of Figure 2G is too small to see clearly; please remove the outer frame of Figure 1D. 

10. Please carefully check the special characters and Spelling in the text, for example:“CtDT toxicity”; “CD63 +”; “NF-kB”; “C jejuni Cytolethal”; “5% CO2”.

11. Please improve and edit your English language and style.

Author Response

In this manuscript, the authors aimed to characterize that the extracellular vesicles from C. jejuni CDT intoxicated-Caco-2 cells for potential application in antitumor strategies. They found that either the wild-type C. jejuni lysate or extracellular vesicles from C. jejuni CDT intoxicated-Caco-2 cells induce cell cycle arrest in G2/M phase whereas this does not take place in CdtA mutant-treated cells. These findings may provide approaches to deliver therapeutic cargoes in anticancer strategies. While the findings are interesting for the field, the manuscript is desired to be revised due to some major and minor concerns as follows: 

  1. For the introduction, it is suggested to divide it into different paragraphs so as to introduce it hierarchically, which will help the readers to grasp the current situation and significance of this study.

Thank you for your suggestion. We have divided the introduction into different paragraphs, introducing differentially the topics of the manuscript.

  1. In lines 93-94, “This study focuses on mitochondrial modifications, deep changes in lysosomal exocytosis, secretory autophagy and EV release.....”. However, in the introduction, the authors did not mention the effects of CDT on these biological processes and its corresponding biological significance.

We have inserted in the introduction the impact of CDT on mitochondria and lysosome network, as detailed in our previously published papers (Canonico et al., 2020, 2018, 2014).

  1. In general, the experimental design in this article is unreasonable:

We followed the overall suggestions and rewrote the manuscript more fluently and specifically indicating the main results and ideas moving the work. We have also reduced the amount of data and figures in order not to lose the main message of the study. We hope now that the experimental design emerges from a simpler, orderly and more fluid text that we have prepared with the support of the valid suggestions of the reviewers.

Finally, we have divided the work into two parts, for more clarity.

  1. jejuni CDT intoxicated-Caco-2 cells should be Caco-2 cells treated with CDT from C. jejuni, not C. jejuni lysate;

We have rewritten the sentences.

  1. CDT toxicity is mainly dependent on CdtB activity, so why use the CdtA mutant as the control?

The reviewer is correct, however the cdtA mutant produces a mutated toxin, unable to enter the cells, since CdtA and CdtC act as carriers to transport the catalytic subunit, CdtB, within host cells. In this way, comparing all data from mutant and wild type toxin, we have the possibility to specifically ascribe the peculiar cell responses to active internalised CDT. At the beginning of our study on C jejuni CDT (Canonico et al., 2014) we had the possibility to employ for this purpose the strain C. jejuni 11168H cdtA mutant and we continued the project with the same strain.

  1. Please provide the details for the strains C. jejuni ATCC 33291 and C. jejuni 11168H CdtA

We have added the details you have referred to.

  1. How to select the dose and time point for jejuni lysate and EVs? We performed several studies on HeLa cells (Canonico et al., 2014) and myeloid cells, both tumour and normal cells (Canonico et al., 2020, 2018). The findings obtained represented the foundation to select the dose and time point for C. jejuni lysates. For EVs administration we based our methods on literature findings (Busatto et al., 2020; Gai et al., 2017).

  1. For NF-κB signaling pathway activation is usually analyzed by detecting its phosphorylation status and transcriptional activity;

Although the reviewer is correct, in the Special Issue of IJMS titled "Flow Cytometry and Its Applications to Molecular Biology and Diagnosis 2.0" we thought it was particularly suitable to apply mAb cytometric detection, which is our forte. Our monoclonal antibody clone 112A1021 is specific for anti-NF-κB p65 (Jia et al., 2016; Khan et al., 2021; Mardjiati et al., 2021).

Other authors applied flow cytometric evaluation for NF-kb (Abohassan et al., 2022; Chen et al., 2020; Maguire et al., 2015; Trapecar et al., 2014):

  1. For Rb protein, its effect on cell proliferation is usually analyzed by detecting its phosphorylation status using Western blot.

Also in this case, the reviewer is correct, however in the Special Issue of IJMS titled "Flow Cytometry and Its Applications to Molecular Biology and Diagnosis 2.0" special issue we thought it was particularly suitable to apply mAb cytometric detection.

Our clone G3-245 (Juan et al., 1998) is specific for pRb total. In our previous study on U937 cells (Canonico et al., 2020) we used flow cytometry and confocal microscopy.

  1. The extracellular vesicles fromjejuni CDT intoxicated-Caco-2 cells inhibit proliferation of tumor intestinal and myeloid cells, is it tumor-specific? Does it kill the normal intestinal and myeloid cells?  

Thank you for these questions. We changed the title of the manuscript, taking into account this consideration. At the time of the study we did not evaluate the normal intestinal and myeloid cells. However, in the article “Monocyte response to different Campylobacter jejuni lysates involves endoplasmic reticulum stress and the lysosomal–mitochondrial axis: When cell death is better than cell survival” (Canonico et al., 2018), we detected a moderate cell death induction (less than 15%) in human normal monocytes, after 48 h of treatment, together with only 30% of decrease in absolute counting.

  1. How to demonstrate the EVs from infected Caco-2 cells contain CDT, and the following phenomenon of cell cycle arrest is caused by CDT uptake?

As performed in previous manuscripts, we compared all findings from treatment by EVs obtained by Caco-2 ATCC conditioning (EvA) to findings from treatment by EVs obtained by Caco-2 mutant conditioning (EvM). We utilised as controls not only the cells without EVs addition but also cells treated with EVs obtained by untreated Caco-2 cells (EvC). Specific CDT effects (cell cycle, blocking, mitochondrial modifications, ROS increase, relevant changes in lysosomal exocytosis, secretory autophagy) were identified and published in our previous manuscripts (Canonico et al., 2020, 2018, 2014).

  1. The identification evidence of extracellular vesicles is insufficient, and please provide the electron microscopic (TEM or SEM) results.

Although TEM is one of the main techniques adopted for EVs identification (as highlighted by the reviewer) it is not the only one. In fact, in a very recent paper (2021) published in International journal of molecular sciences (Khalyfa and Sanz-Rubio, 2021), entitled “The Mystery of Red Blood Cells Extracellular Vesicles in Sleep Apnea with Metabolic Dysfunction”, the characterisation and quantification of EVs was performed through different techniques such as western blots, flow cytometry, and nanoparticle tracking analysis (NTA). Transmission electron microscopy (TEM), scanning EM (SEM), and cryogenic TEM are largely applied to determine EV features (diameters and morphology). EV size distribution and polydispersity in a sample can be also analysed by dynamic light scattering (DLS), which detects the diffusion coefficient of the scattering EVs. Nanoparticle tracking analysis (NTA) allows for the detection of both EV dimensions and concentrations, through the analysis of the EV Brownian motion and the measurement of scattered light or emitted fluorescence. In particular, flow cytometry can be employed for size detection (de Rond et al., 2018; Simeone et al., 2020), particle concentration (Maia et al., 2020) and EV specific markers (Ekström et al., 2022; Sola et al., 2022; Soukup et al., 2022).

FC strategy has an unexplored capability to be applied in the study of populations of EVs, multiplying the number of possible different analysis from reduced volume of sample or from a single biofluid collection.

  1. Keshtvarz et al. produced an engineering of CdtB to reduce immunogenicity and maintain stability as a new drug candidate for tumor therapy. Please discuss the advantages and disadvantages of this study in depth.

We have inserted this reference and we have discussed the advantages and disadvantages of this study.

  1. Line 117 “ATCC treated cells”, what is ATCC?

We have added the requested explanation.

  1. The horizontal and vertical coordinates of the histogram are not standardized, such as “%”.

We are not sure to have fully understood this issue, however we have checked and improved all the figures, taking care to target the horizontal and vertical coordinates.

  1. Please improve your figure legends, for example, what is the meaning of ' ATCC, Mut ' in Figure 1 A?what is the meaning of the Statistical analysis Symbol marked in the figures?

Thank you for the suggestion. We have added the explanation of ATCC and MUT. We have improved the statistical histogram and symbol marked.

  1. Please improve your figures, for example, the text in the picture of Figure 2G is too small to see clearly; please remove the outer frame of Figure 1D. 

We have removed the outer frame of Figure 1D and we have improved the figures.

  1. Please carefully check the special characters and Spelling in the text, for example: “CtDT toxicity”; “CD63 +”; “NF-kB”; “C jejuni Cytolethal”; “5% CO2”.

We carefully checked these special characters.

  1. Please improve and edit your English language and style.

Rewriting the manuscript has allowed us to pay particular attention to the English language and style. We have now overhauled the manuscript significantly.

References

Abohassan, M., Al Shahrani, M., Alshahrani, M.Y., Begum, N., Radhakrishnan, S., Rajagopalan, P., 2022. FNF-12, a novel benzyli-dene-chromanone derivative, attenuates inflammatory response in in vitro and in vivo asthma models mediated by M2-related Th2 cytokines via MAPK and NF-kB signaling. Pharmacol. Reports 74, 96–110. https://doi.org/10.1007/s43440-021-00325-0

Busatto, S., Yang, Y., Walker, S.A., Davidovich, I., Lin, W.H., Lewis-Tuffin, L., Anastasiadis, P.Z., Sarkaria, J., Talmon, Y., Wurtz, G., Wolfram, J., 2020. Brain metastases-derived extracellular vesicles induce binding and aggregation of low-density lipoprotein. J. Nanobiotechnology 18, 1–15. https://doi.org/10.1186/s12951-020-00722-2

Canonico, B., Campana, R., Luchetti, F., Arcangeletti, M., Betti, M., Cesarini, E., Ciacci, C., Vittoria, E., Galli, L., Papa, S., Baffone, W., 2014. Campylobacter jejuni cell lysates differently target mitochondria and lysosomes on HeLa cells. Apoptosis 19, 1225–1242. https://doi.org/10.1007/s10495-014-1005-0

Canonico, B., Cesarini, E., Montanari, M., Di Sario, G., Campana, R., Galluzzi, L., Sola, F., Gundogdu, O., Luchetti, F., Diotallevi, A., Baffone, W., Giordano, A., Papa, S., 2020. Rapamycin re-directs lysosome network, stimulates er-remodeling, involving membrane CD317 and affecting exocytosis, in Campylobacter Jejuni-lysate-infected U937 cells. Int. J. Mol. Sci. 21, 2207. https://doi.org/10.3390/ijms21062207

Canonico, B., di Sario, G., Cesarini, E., Campana, R., Luchetti, F., Zamai, L., Ortolani, C., Nasoni, M.G., Baffone, W., Papa, S., 2018. Monocyte response to different Campylobacter jejuni lysates involves endoplasmic reticulum stress and the lysosomal–mitochondrial axis: When cell death is better than cell survival. Toxins (Basel). 10. https://doi.org/10.3390/toxins10060239

Chen, T., Zhang, X., Zhu, G., Liu, H., Chen, J., Wang, Y., He, X., 2020. Quercetin inhibits TNF-α induced HUVECs apoptosis and inflammation via downregulating NF-kB and AP-1 signaling pathway in vitro. Medicine (Baltimore). 99, e22241. https://doi.org/10.1097/MD.0000000000022241

de Rond, L., Coumans, F.A.W., Nieuwland, R., van Leeuwen, T.G., van der Pol, E., 2018. Deriving Extracellular Vesicle Size From Scatter Intensities Measured by Flow Cytometry. Curr. Protoc. Cytom. 86, e43. https://doi.org/10.1002/CPCY.43

Ekström, K., Crescitelli, R., Pétursson, H.I., Johansson, J., Lässer, C., Bagge, R.O., 2022. Characterization of surface markers on extracellular vesicles isolated from lymphatic exudate from patients with breast cancer. BMC Cancer 22, 1–17. https://doi.org/10.1186/S12885-021-08870-W/FIGURES/5

Gai, C., Gomez, Y., Tetta, C., Felice Brizzi, M., Camussi, G., 2017. Protective Role of Stem Cell Derived Extracellular Vesicles in an In Vitro Model of Hyperglycemia-Induced Endothelial Injury. J. Cell Sci. Ther. 08. https://doi.org/10.4172/2157-7013.1000264

Jia, D., Tan, Y., Liu, H., Ooi, S., Li, L., Wright, K., Bennett, S., Addison, C.L., Wang, L., 2016. Cardamonin reduces chemothera-py-enriched breast cancer stem-like cells in vitro and in vivo. Oncotarget 7, 771–785. https://doi.org/10.18632/ONCOTARGET.5819

Juan, G., Gruenwald, S., Darzynkiewicz, Z., 1998. Phosphorylation of retinoblastoma susceptibility gene protein assayed in indi-vidual lymphocytes during their mitogenic stimulation. Exp. Cell Res. 239, 104–110. https://doi.org/10.1006/excr.1997.3885

Khalyfa, A., Sanz-Rubio, D., 2021. The mystery of red blood cells extracellular vesicles in sleep apnea with metabolic dysfunction. Int. J. Mol. Sci. https://doi.org/10.3390/ijms22094301

Khan, H.U., Aamir, K., Jusuf, P.R., Sethi, G., Sisinthy, S.P., Ghildyal, R., Arya, A., 2021. Lauric acid ameliorates lipopolysaccharide (LPS)-induced liver inflammation by mediating TLR4/MyD88 pathway in Sprague Dawley (SD) rats. Life Sci. 265, 118750. https://doi.org/10.1016/j.lfs.2020.118750

Maguire, O., O’Loughlin, K., Minderman, H., 2015. Simultaneous assessment of NF-κB/p65 phosphorylation and nuclear localiza-tion using imaging flow cytometry. J. Immunol. Methods 423, 3–11. https://doi.org/10.1016/j.jim.2015.03.018

Maia, J., Batista, S., Couto, N., Gregório, A.C., Bodo, C., Elzanowska, J., Strano Moraes, M.C., Costa-Silva, B., 2020. Employing Flow Cytometry to Extracellular Vesicles Sample Microvolume Analysis and Quality Control. Front. Cell Dev. Biol. 8, 1165. https://doi.org/10.3389/FCELL.2020.593750/BIBTEX

Mardjiati, S., Wulan, M., Laswati, H., Hadi, U., Nasronudin, N., 2021. Physical Exercise in Clinical Stage IIhuman Immunodeficiency Virus Infection Patients’ Increasesskeletal Muscle MAss Through the Increasing of Myogenic Regulatory Factors Expression. Indian J. Forensic Med. Toxicol. 15, 2566–2573. https://doi.org/10.37506/ijfmt.v15i4.17091

Simeone, P., Celia, C., Bologna, G., Ercolino, E., Pierdomenico, L., Cilurzo, F., Grande, R., Diomede, F., Vespa, S., Canonico, B., Guescini, M., Stocchi, V., Lotti, L.V., Guagnano, M.T., Stellin, L., Papa, S., Trubiani, O., Marchisio, M., Miscia, S., Lanuti, P., 2020. Diameters and Fluorescence Calibration for Extracellular Vesicle Analyses by Flow Cytometry. Int. J. Mol. Sci. 21, 1–15. https://doi.org/10.3390/IJMS21217885

Sola, F., Montanari, M., Fiorani, M., Barattini, C., Ciacci, C., Burattini, S., Lopez, D., Ventola, A., Zamai, L., Ortolani, C., Papa, S., Canonico, B., 2022. Fluorescent Silica Nanoparticles Targeting Mitochondria: Trafficking in Myeloid Cells and Application as Doxorubicin Delivery System in Breast Cancer Cells. Int. J. Mol. Sci. 23, 3069. https://doi.org/10.3390/ijms23063069

Soukup, J., Kostelanská, M., Kereïche, S., Hujacová, A., Pavelcová, M., Petrák, J., Havrdová, E.K., Holada, K., 2022. Flow Cytometry Analysis of Blood Large Extracellular Vesicles in Patients with Multiple Sclerosis Experiencing Relapse of the Disease. J. Clin. Med. 11, 2832. https://doi.org/10.3390/JCM11102832/S1

Trapecar, M., Goropevsek, A., Gorenjak, M., Gradisnik, L., Rupnik, M.S., 2014. A co-culture model of the developing small intestine offers new insight in the early immunomodulation of enterocytes and macrophages by Lactobacillus spp. through STAT1 and NF-kB p65 translocation. PLoS One 9, 1–8. https://doi.org/10.1371/journal.pone.0086297

Round 2

Reviewer 1 Report

Congratulations. You have done a lot of additional work. Now it's ready for publishing.